# Experiences of people with opioid use disorder during the COVID-19 pandemic: A qualitative study

**Lexis R. Galarneau**[1,2]*, **Jesse Hilburt**[3], **Zoe R. O'Neill**[4], **Jane A. Buxton**[1,5], **Frank X. Scheuermeyer**[6,7], **Kathryn Dong**[8], **Janusz Kaczorowski**[9], **Aaron M. Orkin**[10,11], **Skye Pamela Barbic**[7,12], **Misty Bath**[3], **Jessica Moe**[5,13], **Isabelle Miles**[6], **Dianne Tobin**[14], **Sherry Grier**[15†], **Emma Garrod**[16,17], **Andrew Kestler**[6,7,17]*

**1** School of Population and Public Health, University of British Columbia, Vancouver, British Columbia, Canada, **2** School of Public Health, University of Alberta, Edmonton, Alberta, Canada, **3** Vancouver Coastal Health, Vancouver, British Columbia, Canada, **4** Faculty of Medicine and Health Sciences, McGill University, Montréal, Québec, Canada, **5** British Columbia Centre for Disease Control, Vancouver, British Columbia, Canada, **6** Department of Emergency Medicine, St Paul's Hospital and University of British Columbia, Vancouver, British Columbia, Canada, **7** Center for Health Evaluation and Outcome Sciences, Vancouver, British Columbia, Canada, **8** Department of Emergency Medicine, University of Alberta, Edmonton, Alberta, Canada, **9** Department of Family and Emergency Medicine, University of Montréal, and Centre de Recherche du Centre Hospitalier de l'Université de Montréal, Montréal, Québec, Canada, **10** Department of Family and Community Medicine, University of Toronto, Toronto, Ontario, Canada, **11** Inner City Health Associates, Toronto, Ontario, Canada, **12** Department of Occupational Science and Occupational Therapy, Faculty of Medicine, University of British Columbia, Vancouver, Canada, **13** Department of Emergency Medicine, University of British Columbia, Vancouver, British Columba, Canada, **14** Vancouver Area Network of Drug Users, Vancouver, British Columbia, Canada, **15** Portland Hotel Society Community Services Society, Vancouver, British Columbia, Canada, **16** Providence Health Care, Vancouver, British Columbia, Canada, **17** British Columbia Centre on Substance Use, Vancouver, British Columbia, Canada

† Deceased.

* lexis@ualberta.ca (LG); andrew.kestler@ubc.ca (AK)

**Data Availability Statement:** All relevant data are within the manuscript and its Supporting Information files.

## Abstract

### Aim

To capture pandemic experiences of people with opioid use disorder (OUD) to better inform the programs that serve them.

### Design

We designed, conducted, and analyzed semi-structured qualitative interviews using grounded theory. We conducted interviews until theme saturation was reached and we iteratively developed a codebook of emerging themes. Individuals with lived experience of substance use provided feedback at all steps of the study.

### Setting

We conducted phone or in-person interviews in compliance with physical distancing and public health regulations in outdoor Vancouver parks or well-ventilated indoor spaces between June to September 2020.

**Funding:** The principal investigator (AK) received two grants from separate funders to support this study. The first funder was the Canadian Institutes of Health Research (CIHR) to the Canadian Research Initiative in Substance Misuse (CRISM) (grant number: CRISM Implementation Science Program on Opioid Interventions & Services NRF 154893). The URL of this funder is: https://crism. ca/. The second funder was the Providence Health Care Research Institute and Vancouver Coastal Health Research Institute (grant number: Innovation & Translation Research Award F19-02338). The URLs of this funder is https://www. providenceresearch.ca/ and https://www.vchri.ca/. The funders had no role in study design, data collection and analysis, decision to publish, or preparation of the manuscript.

**Competing interests:** The authors have declared that no competing interests exist.

## Participants

Using purposive sampling, we recruited participants (n = 19) who were individuals with OUD enrolled in an intensive community outreach program, had visited one of two emergency departments, were over 18, lived within catchment, and were not already receiving opioid agonist therapy.

## Measurements

We audio-recorded interviews, which were later transcribed verbatim and checked for accuracy while removing all identifiers. Interviews explored participants' knowledge of COVID-19 and related safety measures, changes to drug use and healthcare services, and community impacts of COVID-19.

## Results

One third of participants were women, approximately two thirds had stable housing, and ages ranged between 23 and 59 years old. Participants were knowledgeable on COVID-19 public health measures. Some participants noted that fear decreased social connection and reluctance to help reverse overdoses; others expressed pride in community cohesion during crisis. Several participants mentioned decreased access to housing, harm reduction, and medical care services. Several participants reported using drugs alone more frequently, consuming different or fewer drugs because of supply shortages, or using more drugs to replace lost activities.

## Conclusion

COVID-19 had profound effects on the social lives, access to services, and risk-taking behaviour of people with opioid use disorder. Pandemic public health measures must include risk mitigation strategies to maintain access to critical opioid-related services.

## Introduction

People who use drugs (PWUD) may be at higher risk of Coronavirus disease 2019 (COVID-19) infection and poorer outcomes due to underlying health conditions [1, 2]. Social inequities faced by PWUD may prevent individuals from adhering to public health measures [1, 3, 4]. A study of over 73 million patients in the USA found that people with opioid use disorder (OUD) had a higher risk of COVID-19 infection compared to people with other substance use disorders [2]. Physical distancing regulations have constrained harm reduction services, such as needle/syringe distribution, further increasing risks for people with OUD [5, 6]. One study from England observed substantial reductions in the operating hours of sites offering needles, and in the number of needles provided to PWUD during April to June 2020 compared to March 2020 [6].

British Columbia (BC) exemplifies the interwoven impacts of concurrent opioid [7, 8] and pandemic emergencies. In 2020, COVID-19 and its ensuing public health measures compounded BC's opioid crisis [5]. Comparing 2020 to 2019, BC reported reductions in attendance for overdose prevention/supervised consumption sites, record high numbers of paramedic calls related to opioid overdoses, and spikes in deaths confirmed or suspected to be

related to illicit drug toxicity [5, 9]. It is important to include people with OUD in decision making processes to better understand how to serve them during public health emergencies.

Using qualitative interviews guided by grounded theory, we aimed to describe the experiences and needs of people with OUD during COVID-19, as identified by participants, to inform public health emergency measures. In particular, we chose to focus on individuals not yet on opioid agonist therapy (OAT), as a marker of those at greatest risk for overdose and short-term mortality. Because OAT and OUD harm reduction interventions have evidence-based life-saving benefits [10–12], we hope our results will be used to better connect people with OUD to needed services during public health emergencies and ultimately decrease morbidity and mortality in this population. Our interviews successfully captured facets of participants' experiences and opinions in their own words.

## Methods

### Study design

We designed, conducted, and analyzed semi-structed interviews using grounded theory, a method of qualitative research using iterative design, purposive sampling, and analysis with constant comparison that ensures themes arise from data and not prior hypotheses [13]. We used the Consolidated Criteria for Reporting Qualitative Research (COREQ) [14].

The University of British Columbia Providence Health Care Research Ethics Board approved the study (H19-02203).

### Research team

Our research team is described according to COREQ requirements [14]. The core research team consisted of six individuals. The interviewers included one female Master of Public Health Student (LG) and one male community outreach worker (JH), the latter being a Master of Social Work Student and having prior working relationships with some of the participants. During intake, JH informed the participants of the intent of the study. Prior to starting the interview, LG discussed the intent of the study in more detail and described the research team while obtaining written informed consent. LG and ZO, a female medical student, served as the primary and secondary coders, respectively. Both coders had no prior experience in qualitative research, but both coders had pre-existing interests in substance use and harm reduction. The principal investigator, a male emergency physician with public health experience (AK), oversaw data collection and coding. SG and DT, both women with lived experience of substance use, contributed to the interview guide design, the data collection processes, the analytic framework, and manuscript drafts by performing face validity checks on themes and providing feedback regarding content and wording at all stages of the study.

### Participant selection and setting

This study was embedded in a larger longitudinal study of patients with OUD offered buprenorphine/naloxone initiation packs and intensive outreach follow-up after visits to one of two emergency departments (EDs) in Vancouver, BC. We excluded those under 18, already receiving OAT, or living outside catchment area. We selected individuals within this larger cohort using purposive sampling, contacting candidates via phone who agreed to further research contact.

### Data collection

We developed an interview guide with input from an expert panel at the Canadian Research Initiative in Substance Misuse (CRISM), local emergency physicians and addiction specialists

(S1 Appendix). The interview guide included follow-up questions and prompts to facilitate comprehensive responses. The interviewers pilot tested the guide with two individuals with lived experience of substance use.

We decided *a priori* to continue interviews until reaching thematic saturation or completing 20 interviews, whichever occurred first. The Vancouver Coastal Health Research Institute approved in-person interviews as part of its COVID-19 research resumption review process. The primary interviewer performed all interviews to maintain consistency. The secondary interviewer (an outreach worker) aided in obtaining written informed consent and providing answers to logistical questions about the study. While awaiting approval for in-person interviewing, we conducted telephone interviews. During interviews, the secondary interviewer left for outreach related questions. While abiding by public health regulations, we performed in-person interviews in parks, in approved outreach offices or participants' residences if a well-ventilated, spacious room was available. The primary interviewer took field notes on observations including body language where appropriate. Information on body language was not coded and did not form part of the data set per se, but rather was recorded in case it might subsequently be needed to clarify any ambiguous answers by participants (e.g. the interviewer recorded if a participant nodded their head or pointed to an object in the interview space). We provided participants with a $50 honorarium and obtained participants' contact information and consent to be contacted for follow-up questions.

## Analysis

We transcribed audio recordings of interviews verbatim using Otter.ai transcription software [15], reviewing all transcripts for accuracy and removing any identifiers. The primary coder used NVivo Pro 12 Software [16] to organize preliminary themes by reading transcripts line by line and grouping answers to open-ended questions. The primary coder quantified answers to closed-ended questions where appropriate. The primary coder sent all transcripts securely to the secondary coder for quality checks on emerging themes. After analyzing the first 12 transcripts, and again after 16 transcripts, the interview guide was revised, and we re-worded some questions to elicit in-depth answers, added questions based on emerging or deviant concepts, and assessed for theme saturation. Theme saturation occurred after 19 interviews.

After transcription and analysis, the two coders iteratively developed a codebook of themes (S1 Table) with additional input from the principal investigator. Two team members with lived experience reviewed the codebook for face validity, providing feedback on content and wording. After the primary coder's final coding, the secondary coder analyzed four random transcripts to check for internal consistency of coding. An individual with lived experience reviewed the final manuscript for content and wording.

## Results

### Interviews and transcripts

We attempted to contact 26 individuals from June to September 2020, of whom 3 did not respond and 4 did not appear for interviews. No individual directly declined to participate. We completed 10 telephone and 9 in-person interviews. Interviews lasted 45 to 90 minutes. We attempted unsuccessfully to contact 1 participant for follow-up questions. One participant received their transcript upon request, but they provided no feedback. While checking for coding internal consistency, the secondary coder agreed with 88% of the primary coder's choices, and during an adjudication meeting, assessed any disagreement as internally consistent and logical.

## Sample characteristics and responses to closed-ended questions

Demographic characteristics at the time of interviews and responses to closed-ended questions are presented in Table 1. Most participants were men and had stable housing. Most reported not falling ill with COVID-19 at the time of interview, and many reported being less likely to access EDs and family doctors during COVID-19.

## Knowledge of COVID-19 and safety measures

Participants expressed varying levels of understanding of COVID-19 and of ways to protect oneself. Several participants reported not caring about or believing in the dangers of COVID-19 and some participants referred to COVID-19 as fake or a scare tactic:

> "I think it's a common cold thing . . . it's a little bit blown out of proportion." (Participant 10; male, 40 years old)

> "I think it's just a ruse like most major problems are that are global." (Participant 18; male, 34 years old)

Some participants demonstrated considerable COVID-19 knowledge, including its low mortality rate, high transmissibility, and severity based on prior health status.

When asked how participants first heard about COVID-19 and how to stay safe, over half cited the media (e.g. news, television) while others mentioned friends/family, word on the street, as well as housing, harm reduction and treatment centers. Some participants mentioned ubiquity of COVID-19 information and related public health measures:

> "Our society was saturated with it. It was saturated with COVID information. I think it would be impossible to not have been aware and of what was happening and know what to do." (Participant 2; male, 48 years old)

All participants were aware of multiple public health measures including wearing masks, physical distancing, avoiding touching one's face, and frequently sanitizing hands and surfaces. Some participants reported neither following public health measures nor changing their behaviours during COVID-19.

## Emotions during COVID-19

Participants frequently mentioned fear, either of the virus itself or of the overall pandemic impact:

> "I just remember a lot of panic . . . a lot of panic." (Participant 14; female, 28 years old)

> "When everything was shut down, that was . . . terrible. That was terrifying. That scared me to see your whole entire city just like go dark." (Participant 12; female, 37 years old)

Over half of participants believed the public was scared to help someone having an overdose or to help in other ways (Table 1). Some participants specified that fear of helping others was greater at the beginning of the pandemic. A few did not believe fear affected overdose response willingness, reporting that the need to respond to an overdose outweighed potential COVID-19 transmission and feeling pride in the community's willingness to help:

**Table 1. Baseline characteristics of participants reported at the time of interview and frequency counts of answers to closed-ended questions.**

| Characteristic/Answer | n (%) |
|---|---|
| Gender | |
| Male | 13/19 (68%) |
| Female | 6/19 (32%) |
| Age (Years) | |
| Median Age | 34 |
| Age Range | 23–59 |
| Housing Situation | |
| Housed | 12/19 (63%) |
| Not Housed (i.e. no fixed address, sheltered, transitioning into housing, housed in treatment) | 7/19 (37%) |
| Have been sick with COVID-19 | |
| Yes | 0/19 (0%) |
| No | 18/19 (95%) |
| Maybe (was not tested at time of sickness) | 1/19 (5%) |
| Have gotten tested for COVID-19 | |
| Yes | 6/18 (33%) |
| No | 12/18 (67%) |
| Do you think people were scared to help someone having an overdose? | |
| Yes | 13/19 (68%) |
| No | 3/19 (16%) |
| I Don't Know | 3/19 (16%) |
| Do you think people were scared to help people in other ways? | |
| Yes | 15/19 (79%) |
| No | 3/19 (16%) |
| I Don't Know | 1/19 (5%) |
| Did COVID-19 change your ability to you drugs as safely as possible? | |
| Yes, less safe | 9/18 (50%) |
| Yes, more safe | 1/18 (6%) |
| No Change | 8/18 (44%) |
| Did the social or physical distancing recommendations make you use alone or take other risks you might not usually take? | |
| Yes | 7/18 (39%) |
| No | 11/18 (61%) |
| Did you find you were using more or less drugs during COVID-19? | |
| More | 9/19 (47%) |
| Less | 2/19 (11%) |
| No Change | 5/19 (26%) |
| Depends on the kind | 3/19 (16%) |
| Did you find you were buying more or less drugs at a time? | |
| More | 5/10 (50%) |
| Less | 1/10 (10%) |
| No Change | 3/10 (30%) |
| Depends on the kind | 1/10 (10%) |
| Did you find you were using different kinds of drugs during COVID-19? | |
| Yes | 8/18 (44%) |
| No | 9/18 (50%) |
| I Don't Know | 1/18 (6%) |

(*Continued*)

**Table 1.** (Continued)

| Characteristic/Answer | n (%) |
|---|---|
| Did COVID-19 affect your ability to get medical care if you needed it? | |
| Yes | 9/19 (47%) |
| No | 10/19 (53%) |
| Did COVID-19 affect your ability to get the social support services you may have needed? | |
| Yes | 7/17 (41%) |
| No | 10/17 (59%) |
| Did you need a shelter during COVID-19? | |
| Yes, and received one | 5/19 (26%) |
| Yes, but did not receive one | 1/19 (5%) |
| No | 13/19 (68%) |
| Did COVID-19 make it more or less likely you would go to the emergency department? | |
| More | 2/19 (11%) |
| Less | 8/19 (42%) |
| No Change | 9/19 (47%) |
| Did COVID-19 make it more or less likely you would go to your regular doctor or clinic? | |
| More | 1/17 (6%) |
| Less | 7/17 (41%) |
| No Change | 9/17 (53%) |
| Did COVID-19 make it more or less likely you would call 911? | |
| More | 1/17 (6%) |
| Less | 2/17 (12%) |
| No Change | 14/17 (82%) |

"I don't think it really jumps into play. I think it's really out of . . . out of your mind when you see somebody overdosing. I think that the need for helping the person and the general care for that person's wellbeing supersedes it." (Participant 10; male, 40 years old)

"I am quite proud of people . . . if someone's overdosing people still stepped up and saved them." (Participant 2; male, 48 years old)

Anxiety and stress were also frequently reported emotions, related to both the risk of infection and to overall impacts of COVID-19. Other emotions included uncertainty, isolation, frustration, and being tired of COVID-19:

"I don't know how I feel about [COVID-19]. I feel like I'm left in the dark about it, because I don't have a cell phone. I have no access to internet at the moment. So, I really don't know what's going on." (Participant 13; female, 33 years old)

"People are tired of it, and they just want things to get back to normal and there's less of an acceptance of that this is the new normal." (Participant 2; male, 48 years old)

## Drug use and safety during COVID-19

About half of participants reported using drugs less safely during COVID-19, less than half reported no change and one participant reported using drugs *more* safely (Table 1). Of those who reported using drugs less safely, participants cited COVID-19 measures such as building closures, limited capacity at harm reduction sites and physical distancing:

"Overdose prevention sites weren't seeing nearly as many, only half capacity." (Participant 3; male, 40 years old)

"Because we had to be by ourselves, we weren't allowed to be around each other. So basically, forcing us to use alone" (Participant 8; female, 30 years old)

One participant explained that although nearby overdose prevention sites remained open, limited capacity and long waits deterred use:

"[Overdose prevention sites] only allow like a certain amount of people in. There's no waiting room inside. I'm not saying that everything was perfect the way it was before, but it was definitely more likely for me to go in and use that place that service then, then is now. I don't even bother." (Participant 12; female, 37 years old)

Of those reporting a decrease in drug use safety, many agreed that physical distancing recommendations had led them to use alone or to take other risks they normally would not take. Some participants noted that COVID-19 did not impact their drug use:

"I still didn't use alone, like it didn't affect my drug use at all from COVID." (Participant 2; male, 48 years old)

"Protocol has been the same." (Participant 15; male, 31 years old)

A few participants mentioned new strategies during COVID-19 to use drugs *more* safely, including using a virtual supervised consumption app while using alone, not sharing supplies as frequently, and maintaining hygienic spaces:

"I think the one thing that [COVID-19] did do was maybe lift awareness around the importance of maintaining a clean space." (Participant 10; male, 40 years old)

About half the participants reported using more drugs during COVID-19, a few reported using less drugs, some reported no change, and some explained it depended on the kind of drug (Table 1). Two participants reported using more drugs due to boredom:

"If I'm homeless on the street, I'm probably going to be using drugs." (Participant 1; male, 31 years old)

One participant explained that although personal consumption remained stable, the pandemic triggered an urge to use more. Purchasing habits of substances also varied during COVID-19 (Table 1). Some participants bought additional drugs each transaction to limit in-person contact. Another participant mentioned government funding as a cause of increasing purchasing. Almost half of participants reported using different kinds of drugs, with reasons including border closures and changes in drug availability and prices:

"[COVID-19 is] when I started to use the, like benzos mixed in with the fentanyl." (Participant 12; female, 37 years old)

"Less access to the crystal meth and speed because of all of it. So now people have to pay more, and we can't afford it. So, going with out speed, which has been hard." (Participant 13; female, 33 years old)

Four participants reported being contacted by health providers about pharmaceutical alternatives and all four were able to receive a supply. (Pharmaceutical alternatives are substances that can be prescribed to PWUD to offer similar effects as the toxic drugs they are accustomed to but with greater safety [17].) Other participants reported hearing about alternatives elsewhere (e.g. from friends), with some later seeking and receiving a supply. One participant stated that the offered alternative was not amenable to a preferred method of administration, while other participants found the supply of alternatives helpful:

> "Nobody that kind of reached out, but it is something . . . that I have taken advantage of, and I think it's a very positive thing and helped a lot of people in the community. I think it's positive, I think it's a positive thing." (Participant 10; male, 40 years old)

### Interactions with care services during COVID-19

Almost half of participants reported difficulty accessing medical services and several participants reported difficulty accessing social support services (Table 1). Such services had uncertain availability, took longer to access, or were closed:

> "I could never reach anybody and . . . I just assumed that everything was closed." (Participant 7; male, 35 years old)

> "I don't even know if [social support services] were available." (Participant 1; male, 31 years old)

Many participants had difficulty accessing housing or shelters with some shelters not accepting anyone:

> "I've been trying to find housing the whole time, and also by going out and going to house viewings and going around to the different shelters and I've been doing that this whole time and still there's like no housing." (Participant 16; female, 25 years old)

> "I do need a place to stay right now. But um, I was told that a lot of the shelters aren't taking anybody. That's what I was told but may have changed now." (Participant 9; male, 23 years old)

One participant discussed the importance of housing during emergencies, stating that if self-isolation is a recommended strategy then "helping people get housed is really important" (Participant 14; female, 28 years old).

A few participants commented on a positive change in ED care during COVID-19, including fewer crowds or faster and kinder service:

> "I find there's more help at the counter . . . I was called up and I was called up pretty quickly." (Participant 17; male, 45 years old)

Other participants said the ED became slower and more unwelcoming (e.g. staff appearing more judgemental or condescending). Some participants noted that many services had become less user-friendly due to COVID-19 measures. One participant cited new hospital restrictions on visitation while others mentioned reduced capacities of EDs, the need to sanitize hands frequently and wear masks, and physical distancing during appointments or outreach contact.

"... during COVID not being allowed to go in with my partner, you know, when he overdosed." (Participant 13; female, 33 years old)

"[Emergency departments] can't attend to people as much ... because of all the cleaning 'procedures and behind glass and stuff." (Participant 17; male, 45 years old)

One participant conceded that EDs had to prioritize care, which led to some patients being ignored:

"I guess they just only more paid attention to the most serious stuff like whatever seems less important was kind of just you know, put to the side." (Participant 13; female, 33 years old)

Another participant expressed frustration for being placed in the ED next to COVID-19 patients, despite not having COVID-19 symptoms. Some participants cited preference for telephone appointments with care providers compared to their usual in-person appointments. One participant reported the outreach team visited less frequently during COVID-19. Some participants reported being less likely to access care services (Table 1) while others reported no difficulty or changes in accessing services:

"They're still able to get everything I needed. The people really stepped up for that. It was pretty seamless." (Participant 2; male, 48 years old)

## Community impact of COVID-19

Participants explained how their community became empty or like a 'ghost town' during COVID-19:

"I used to walk down [Vancouver street] at night and like it'd be like everybody would be out at the bar, but it's like, non-existent anymore." (Participant 9; male, 23 years old)

Many participants noticed increased tension in personal interactions, describing people as "more distanced and cold[er]" (Participant 13; female, 33 years old) or "closed off" (Participant 12; female, 37 years old). Participants discussed difficulty providing for themselves financially, including job loss, decreased public donations, and greater challenges selling items:

"Yeah, they're scared to donate now. And they're scared to buy things from us. Like, I don't know, the way that some people support themselves, they, you know, sell things and that whole industry is collapsed basically, for those people." (Participant 13; female, 33 years old)

Some participants were concerned about increasing crime such as theft and difficulty identifying individuals with masks on. Others believed physical distancing and other behaviour changes were less pronounced in their community compared to surrounding communities. One participant described how COVID-19 highlighted the importance of social connection:

"I learned a lot more about like how much people matter and the quality of interaction with people, being able to voice yourself your opinions ... when you're able to listen to somebody ... and connect with that person." (Participant 10; male, 40 years old)

## Discussion

We collected opinions of people with OUD to better inform programs that serve them during COVID-19. Participant involvement was very effective and influenced the study by the depth of experiences and opinions shared. Predictably, many participants had difficulty accessing medical, social support or harm reduction services during COVID-19. Most participants reported changing their behaviours to reduce the risk of COVID-19 infection, including wearing masks, sanitizing frequently and physical distancing. Many participants described purchasing more or different kinds of substances. Public health measures such as physical distancing reduced accessibility and use of overdose prevention sites and increased how often individuals use alone.

Our results are similar to COVID-19-related changes in drug use and outcomes reported elsewhere [5, 6, 18–20]. Some participants discussed changes in the kind of substances they use and their purchasing habits using during COVID-19 due to border closures, inability to access certain drugs, or changes in drug prices, which was previously anticipated [5]. Many participants accessed supervised consumption and overdose prevention sites less frequently due to closures or long wait times as a result of physical distancing requirements, consistent with prior research [5, 6, 19, 20]. Our results suggest that physical distancing recommendations and the inability to access safe spaces has created situations where individuals used drugs less safely. The related experiences of using drugs with fewer safeguards and using more or different kinds of drugs is consistent with the increases in overdose-related paramedic calls and mortality during COVID-19 [5, 15, 16]. Several participants also discussed reduced access to housing during COVID-19, consistent with previous research among PWUD [20].

Although our interviews focused on COVID-19 specifically, one can infer that similar disruptions may occur to the lives and accessibility to services of people with OUD during other public health emergencies, whether related to communicable diseases or to other natural disasters. Our findings and recommendations may therefore be useful not only in the near future during the remainder of COVID-19, but also beneficial in other public health emergencies to continue to support people with OUD. First, we recommend expanding access to harm reduction services during public health emergencies. Given physical distancing requirements, we recommend increasing the number of harm reduction sites or adapting safe services protocols to maintain overall capacity. Although this strategy would not benefit those who have access to harm reduction services but choose not to use them due to perceived infection risks, implementing new distribution methods of safer use supplies such as home delivery, vending machines [21], or peer-supported distribution systems have been suggested to mitigate COVID-19 impacts [6]. Many participants noted greater emotional suffering during COVID-19, consistent with previous research [22]. It would certainly benefit people with OUD to have maintained or even enhanced access to mental health care during periods of high uncertainty and anxiety. Since participants additionally reported difficulty accessing other social services, such as housing support during the pandemic, we also recommend maintaining access to all social supports during public health emergencies. Given that some participants preferred virtual appointments during a pandemic, a theme also observed in previous studies [20, 22], we believe virtual care and other alternative methods to connect with services should be expanded and maintained for people with OUD even outside of public health emergencies. Examples of virtual health that may be beneficial include over the phone appointments with care providers to improve access to healthcare or promoting use of virtual supervised consumption apps or national overdose hotlines for individuals to use safely when access to harm reduction services may be limited. However, the opportunities for in-person care should not be sacrificed when introducing virtual services since many people with OUD still prefer face-to-face services, lack

the access to and facility with technology needed for virtual care, or believe virtual care reduces the quality of care provided [22]. Losing access to these services may pose a greater health risk to PWUD than those of the public health emergency itself. Overall, a more flexible healthcare system will be more accessible to people with OUD. As only 33% of participants reported being tested for COVID-19, our results suggest some avoidance of pandemic public health measures services among people with OUD. Conversely, other participants reported very conscientious adherence to public health recommendations but avoided previously utilized OUD-related services. We therefore recommend coordinating the provision of public health measurers and opioid use services, such as disseminating harm reduction information or supplies at COVID-19 testing or vaccination centres, and offering COVID-19 testing and vaccination at harm reduction outlets or alongside outreach visits. We believe intertwining interventions for multiple public health emergencies, in this case COVID-19 and the opioid overdose crisis, will improve access and coverage across the board. We also recommend increasing information about public health emergencies tailored to people with OUD. For example, many regions created COVID-19 guides specifically for PWUD [23–25]. Having information directed and tailored to people with OUD may help inform people about opioid-specific safety measures during public health emergencies. Finally, we must note that our recommendations and the expansions of services for people with OUD should not be exclusive to during COVID-19 or the next public health emergency. The ongoing opioid emergency calls for increased support regardless of the status of concurrent emergencies. Resources initially devoted to COVID-19 and deemed no longer necessary should be shifted to expand access to harm reduction services and augment care services for people with OUD.

## Limitations

We recruited participants from Vancouver EDs who were over 18 and not receiving OAT at intake, thus our results may not reflect experiences of people with OUD in other settings. Further research may be required to demonstrate consistency in other settings. The span of sociodemographic variables in our sample may also limit generalizability. Our participants had agreed to offer opinions on EDs and outreach services, which may further contribute to selection bias. Social desirability bias is a concern for any qualitative interview assessment. Participants were not involved in any feedback on results; this was mitigated by having individuals with lived experience perform face validity checks. Critically, we conducted interviews from June to September 2020, with low local COVID-19 incidence but prior to widespread vaccination; a change in either of these features would likely influence public health measures and the impact on respondents. Since the interviews, opinions and experiences may have changed; future research is required to demonstrate any changes or preservation of findings.

## Conclusion

Public health measures limit access to medical and social support services and thus directly affect the lives of people with OUD. Decision makers should mitigate the impacts on people with OUD by meaningfully involving them when implementing emergency measures, by expanding access to harm reduction, and by maintaining or augmenting pre-existing services.

## Supporting information

**S1 Table. Thematic codebook.** Thematic codebook used for the qualitative analysis of participants' opinions and experiences during COVID-19 gathered in interviews.
(DOCX)

**S1 Appendix. Interview guide.** Interview guide used to gather participants' opinions and experiences during COVID-19.
(DOCX)

**S1 File. Participant transcripts.** Participant transcripts transcribed verbatim from audio recordings with all names or potentially identifying information removed. Unless otherwise specified, "Interviewer" refers to the primary interviewer (LG).
(DOCX)

## Acknowledgments

Dr Scheuermeyer gratefully recognizes support from the British Columbia Emergency Medicine Network.

## Author Contributions

**Conceptualization:** Lexis R. Galarneau, Jane A. Buxton, Frank X. Scheuermeyer, Kathryn Dong, Janusz Kaczorowski, Aaron M. Orkin, Misty Bath, Jessica Moe, Isabelle Miles, Dianne Tobin, Sherry Grier, Emma Garrod, Andrew Kestler.

**Formal analysis:** Lexis R. Galarneau, Jesse Hilburt, Zoe R. O'Neill, Jane A. Buxton, Frank X. Scheuermeyer, Skye Pamela Barbic, Dianne Tobin, Sherry Grier.

**Funding acquisition:** Janusz Kaczorowski, Andrew Kestler.

**Investigation:** Lexis R. Galarneau, Jesse Hilburt, Zoe R. O'Neill, Dianne Tobin, Sherry Grier, Andrew Kestler.

**Methodology:** Lexis R. Galarneau, Jane A. Buxton, Frank X. Scheuermeyer, Kathryn Dong, Janusz Kaczorowski, Aaron M. Orkin, Skye Pamela Barbic, Misty Bath, Jessica Moe, Isabelle Miles, Dianne Tobin, Sherry Grier, Emma Garrod, Andrew Kestler.

**Project administration:** Lexis R. Galarneau, Jesse Hilburt, Misty Bath, Andrew Kestler.

**Supervision:** Andrew Kestler.

**Writing – original draft:** Lexis R. Galarneau, Andrew Kestler.

**Writing – review & editing:** Lexis R. Galarneau, Jesse Hilburt, Zoe R. O'Neill, Jane A. Buxton, Frank X. Scheuermeyer, Kathryn Dong, Janusz Kaczorowski, Aaron M. Orkin, Skye Pamela Barbic, Misty Bath, Jessica Moe, Isabelle Miles, Dianne Tobin, Sherry Grier, Emma Garrod, Andrew Kestler.

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
