## [Decision Letter · Decision Letter 0]

29 Jun 2021

PONE-D-21-16945

Experiences of people with opioid use disorder during the COVID-19 pandemic: a qualitative study

PLOS ONE

Dear Dr. Galarneau,

Thank you for submitting your manuscript to PLOS ONE. After careful consideration, we feel that it has merit but does not fully meet PLOS ONE’s publication criteria as it currently stands. Therefore, we invite you to submit a revised version of the manuscript that addresses the points raised during the review process.

We look forward to receiving your revised manuscript.

Kind regards,

Christine Leong, Pharm. D.

Academic Editor

PLOS ONE

Journal Requirements:

Reviewers' comments:

Reviewer's Responses to Questions

**Comments to the Author**

1. Is the manuscript technically sound, and do the data support the conclusions?

Reviewer #1: Yes

Reviewer #2: Yes

2. Has the statistical analysis been performed appropriately and rigorously? 

Reviewer #1: N/A

Reviewer #2: Yes

3. Have the authors made all data underlying the findings in their manuscript fully available?

Reviewer #1: Yes

Reviewer #2: Yes

4. Is the manuscript presented in an intelligible fashion and written in standard English?

Reviewer #1: Yes

Reviewer #2: Yes

5. Review Comments to the Author

Reviewer #1: Concise and well-written manuscript overall, methods are clear, results correspond well with existing other data including an increase in opioid overdose and mortality, but provide an important human perspective into the impacts of COVID on PWUD.

The recommendations in the final discussion paragraph are great, however, even if they were implemented immediately upon publication, would probably come too late. One suggestion would be to rework this content to call for investment in emergency preparedness with respect to the needs of PWUD or harm reduction and addiction services generally, whether for another pandemic, natural disaster, etc.

Reviewer #2: Dear Lexis Galarneau et. al.,

Many thanks to you and your team for conducting valuable qualitative research on the subject of pressing and intersecting public health crises – COVID-19 and opioid overdose related morbidity and mortality. Overall, the manuscript was clear and concise in presenting the study’s background, methodology, results, discussion and conclusion.

Please see below for specific comments and questions:

- Curious as to how the information regarding body language during the interviews was used

- Given the positive feedback regarding alternative ways to connect (i.e., virtual appointments), wonder what authors thought about this as a way to improve access to services

- Lines 290, 294, 295 (and more) – would be worth qualifying what “a few”, “ other participants, “some participants” actually represents

- Unclear how the data collected directly corresponds to recommendations.

o Providing harm reduction information at COVID-19 testing and vaccination centres seems like a sound strategy but how does the data in the manuscript support this? Approximately 33% of the participants had been tested for COVID-19.

o For the theme of ‘emotions during COVID-19’, any thoughts on how to address this? Improved access to mental health services?

o Given that the group, by definition, has OUD and is not currently receiving OAT, I would assume this would’ve been a clear, evidence-based strategy to decrease morbidity and mortality

For your information, I have included a Canadian study with similar intent, methodology and results for your review: https://www.journalofsubstanceabusetreatment.com/article/S0740-5472(21)00100-8/fulltext

6. PLOS authors have the option to publish the peer review history of their article (what does this mean?). If published, this will include your full peer review and any attached files.

Reviewer #1: **Yes: **Michael A. Beazely

Reviewer #2: No

---

## [Author Response · Author response to Decision Letter 0]

14 Jul 2021

Response to Editors’ comments

Comment 1: Please ensure that your manuscript meets PLOS ONE's style requirements, including those for file naming. The PLOS ONE style templates can be found at https://journals.plos.org/plosone/s/file?id=wjVg/PLOSOne_formatting_sample_main_body.pdf and https://journals.plos.org/plosone/s/file?id=ba62/PLOSOne_formatting_sample_title_authors_affiliations.pdf

Response: The manuscript has been revised to comply with all style requirements, including editing font sizes and changing file names. 

Comment 2: Please review your reference list to ensure that it is complete and correct. If you have cited papers that have been retracted, please include the rationale for doing so in the manuscript text, or remove these references and replace them with relevant current references. Any changes to the reference list should be mentioned in the rebuttal letter that accompanies your revised manuscript. If you need to cite a retracted article, indicate the article’s retracted status in the References list and also include a citation and full reference for the retraction notice. 

Response: All references have been revised to ensure completeness and correctness. Reference style has been revised to match the Vancouver style requested by PLOS ONE. All links have been checked to ensure they are still working and correct, and no cited papers haven been retracted. 

Comment 3: We note that you have indicated that data from this study are available upon request. PLOS only allows data to be available upon request if there are legal or ethical restrictions on sharing data publicly. For information on unacceptable data access restrictions, please see http://journals.plos.org/plosone/s/data-availability#loc-unacceptable-data-access-restrictions.

Response: At the time of our original submission, we had some concerns about the participants’ privacy and had opted to make interview transcripts only available upon request. However, after further review and removal of a very few potentially identifying details in the transcripts, we now feel the transcript are sufficiently anonymized to permit broader sharing. We have consolidated all transcripts into one Supporting Information file, as suggested. We have named this document “S1 Transcripts” and have submitted it along-side our revised manuscript.

Response to Reviewer 1 comments

Reviewer 1 comments:

Concise and well-written manuscript overall, methods are clear, results correspond well with existing other data including an increase in opioid overdose and mortality, but provide an important human perspective into the impacts of COVID on PWUD.

The recommendations in the final discussion paragraph are great, however, even if they were implemented immediately upon publication, would probably come too late. One suggestion would be to rework this content to call for investment in emergency preparedness with respect to the needs of PWUD or harm reduction and addiction services generally, whether for another pandemic, natural disaster, etc.

Response: We agree with the reviewer that our recommendations would be most impactful if written through lens of public health emergencies as a whole, including the current opioid crisis, rather than through the narrower perspective of COVID-19 alone. We thank the reviewer for bringing this to our attention. We have revised our wording and edited our discussion to speak to measures implemented in public health emergencies more generally.

Response to Reviewer 2 comments

Reviewer 2 comments:

Many thanks to you and your team for conducting valuable qualitative research on the subject of pressing and intersecting public health crises – COVID-19 and opioid overdose related morbidity and mortality. Overall, the manuscript was clear and concise in presenting the study’s background, methodology, results, discussion and conclusion.

Please see below for specific comments and questions:

- Curious as to how the information regarding body language during the interviews was used

Response: In retrospect, our description in the methods may have been misleading. The interviewer recorded body language information not to form part of the study data set directly nor code it during analysis, but rather to help the interviewer clarify potentially ambiguous answers from the participants, i.e. as a prompt for further clarifying questions. For example, it would be recorded if a participant nodded her head or pointed to something in the interview space. We thank the reviewer for bringing this to our attention, and to ensure clarity for future readers we have added a description of how body language information was used in our methods section.

- Given the positive feedback regarding alternative ways to connect (i.e., virtual appointments), wonder what authors thought about this as a way to improve access to services

Response: We strongly agree with the reviewer that alternative ways to connect such as virtual appointments could greatly improve access to care for some people with OUD. We also believe these alternative means could be beneficial in non-emergency situations as some of our participants expressed preference for virtual appointments and appreciated not having to find transportation to go see their doctors. We thank the reviewer for this suggestion and have included alternative ways to connect as a recommendation in our discussion. 

- Lines 290, 294, 295 (and more) – would be worth qualifying what “a few”, “ other participants, “some participants” actually represents

Response: We agree with the reviewer that numbers may help some readers quantify our results, however, we intentionally chose to exclude numbers from our textual results section as to not distract from the qualitative nature of our study. That being said, for readers who wish to see numeric data, we have included numbers in Table 1 (located under the results section entitled “Sample characteristics and responses to closed-ended questions”), as well as have included the counts (i.e. number of references) of all themes in our codebook, which has been attached as a supplementary information file entitled “T1 Table”. As further justification for our approach, we reference as an example a qualitive study published in PLOS ONE in 2020. This study’s authors also avoided including numbers in their text, using terms like “nearly all”, “many participants”, “other participants”, and “some participants”. Article link: https://journals.plos.org/plosone/article?id=10.1371/journal.pone.0230408

- Unclear how the data collected directly corresponds to recommendations.

o Providing harm reduction information at COVID-19 testing and vaccination centres seems like a sound strategy but how does the data in the manuscript support this? Approximately 33% of the participants had been tested for COVID-19.

Response: We agree with the reviewer that the link between our data and our recommendations could be strengthened in this instance. Indeed, the fact that only 33% of participants reported having been tested for COVID-19 suggests that some participants may have been reluctant to interact with public health services, including testing. That being said, many participants also voiced accurate understanding of the COVID-19 public health measures and a desire to adhere to them. We believe the greatest danger lies in siloed approaches to 2 public health crises, the opioid epidemic crisis and COVID-19. Those most adherent to COVID regulations may have accessed COVID-19 services such as testing but self-isolated to the point of avoiding overdose prevention sites and other harm reduction services. Those least concerned with COVID may have continued using harm reduction services as similarly as possible to their pre-pandemic patterns. Therefore, we recommend that greater integration of programs and services across public health priorities would benefit people with OUD. For example, people with OUD could receive COVID-19 testing at harm reduction facilities, or receive harm reduction supplies at vaccination and testing sites. This reasoning concept has now been more clearly stated in the Discussion to ensure clarity for future readers. 

o For the theme of ‘emotions during COVID-19’, any thoughts on how to address this? Improved access to mental health services?

Response: As participants reported emotional suffering during COVID-19, we agree with the reviewer’s recommendation to strengthen mental health supports during public health emergencies. The reviewer’s comment also brought to our attention that participants had difficulty accessing other social supports, such as housing, and these supports should be maintained during public health emergencies as well. We have revised our discussion to include a recommendation for mental heath and other social supports during public health emergencies. We thank the reviewer for the suggestion. 

o Given that the group, by definition, has OUD and is not currently receiving OAT, I would assume this would’ve been a clear, evidence-based strategy to decrease morbidity and mortality

Response: We give thanks to the reviewer for pointing out that we, by design, specifically focused on individuals not yet on OAT. We have now made this clearer at the end of the introduction, provided a rationale for doing so, and included a reference on the evidence-base of OAT benefit. 

For your information, I have included a Canadian study with similar intent, methodology and results for your review: https://www.journalofsubstanceabusetreatment.com/article/S0740-5472(21)00100-8/fulltext

Response: We thank the reviewer for bringing this article to our attention. This article was not available at first manuscript draft, and we agree this study’s results are a valuable addition to our discussion. We have revised our discussion and references to include this study to help strengthen our manuscript.

---

## [Editor Report · Decision Letter 1]

16 Jul 2021

Experiences of people with opioid use disorder during the COVID-19 pandemic: a qualitative study

PONE-D-21-16945R1

Dear Dr. Galarneau,

We’re pleased to inform you that your manuscript has been judged scientifically suitable for publication and will be formally accepted for publication once it meets all outstanding technical requirements.

Kind regards,

Christine Leong, Pharm. D.

Academic Editor

PLOS ONE
---

## [Editor Report · Acceptance letter]

22 Jul 2021

PONE-D-21-16945R1 

Experiences of people with opioid use disorder during the COVID-19 pandemic: a qualitative study 

Dear Dr. Galarneau:

I'm pleased to inform you that your manuscript has been deemed suitable for publication in PLOS ONE. Congratulations! Your manuscript is now with our production department. 

Kind regards, 

on behalf of

Dr. Christine Leong 

Academic Editor

PLOS ONE